# The unquantified mass loss of Northern Hemisphere marine-terminating glaciers from 2000–2020

William Kochtitzky [1] ✉, Luke Copland[1], Wesley Van Wychen [1,2], Romain Hugonnet [3,4,5], Regine Hock[6,7], Julian A. Dowdeswell [8], Toby Benham [8], Tazio Strozzi [9], Andrey Glazovsky[10], Ivan Lavrentiev[10], David R. Rounce[11], Romain Millan[12], Alison Cook [1], Abigail Dalton[1], Hester Jiskoot [13], Jade Cooley[13], Jacek Jania [14] & Francisco Navarro [15]

In the Northern Hemisphere, ~1500 glaciers, accounting for 28% of glacierized area outside the Greenland Ice Sheet, terminate in the ocean. Glacier mass loss at their ice-ocean interface, known as frontal ablation, has not yet been comprehensively quantified. Here, we estimate decadal frontal ablation from measurements of ice discharge and terminus position change from 2000 to 2020. We bias-correct and cross-validate estimates and uncertainties using independent sources. Frontal ablation of marine-terminating glaciers contributed an average of $44.47 \pm 6.23$ Gt a$^{-1}$ of ice to the ocean from 2000 to 2010, and $51.98 \pm 4.62$ Gt a$^{-1}$ from 2010 to 2020. Ice discharge from 2000 to 2020 was equivalent to $2.10 \pm 0.22$ mm of sea-level rise and comprised approximately 79% of frontal ablation, with the remainder from terminus retreat. Near-coastal areas most impacted include Austfonna, Svalbard, and central Severnaya Zemlya, the Russian Arctic, and a few Alaskan fjords.

When glaciers terminate in the ocean, mass is lost by frontal ablation at the ice-ocean interface, including mass loss due to iceberg calving (i.e., the mechanical release of icebergs or smaller pieces of ice to the ocean), submarine frontal melting, and subaerial frontal melting and sublimation[1]. Quantifying frontal ablation is essential to partitioning glacier mass loss components, which informs sea-level budgets, estimates of glacier freshwater contributions to the ocean, impacts on marine ecosystems and iceberg hazards. Recent model results suggest that Northern Hemisphere frontal ablation was around 39 Gt a$^{-1}$ from

1980 to 1999[2]. Frontal ablation accounted for 10% of the 882 Gt a$^{-1}$ global glacier ablation rate (which includes surface melt), although about half of this occurred in the Southern Hemisphere[2]. The same modeling study projected a Northern Hemisphere frontal ablation rate of $50.6 \pm 23.8$ Gt a$^{-1}$ for 2020–2040[2]. However, observation-based estimates of frontal ablation have been limited, partially because many studies only quantify ice discharge[3,4] as they do not include mass changes due to terminus retreat or advance, or these are only computed for individual glaciers[5]. Alaska[6] and Svalbard[7] are the only

[1]Department of Geography, Environment and Geomatics, University of Ottawa, Ottawa, Ontario K1N 6N5, Canada. [2]Department of Geography and Environmental Management, University of Waterloo, Waterloo, Ontario, Canada. [3]LEGOS, Université de Toulouse, CNES, CNRS, IRD, UPS, Toulouse, France. [4]Laboratory of Hydraulics, Hydrology and Glaciology (VAW), ETH Zürich, Zürich, Switzerland. [5]Swiss Federal Institute for Forest, Snow and Landscape Research (WSL), Birmensdorf, Switzerland. [6]Department of Geosciences, University of Oslo, Oslo, Norway. [7]Geophysical Institute, University of Alaska Fairbanks, Fairbanks, USA. [8]Scott Polar Research Institute, University of Cambridge, Cambridge, UK. [9]Gamma Remote Sensing, Gümligen (BE), Switzerland. [10]Institute of Geography, Russian Academy of Sciences, Moscow, Russia. [11]Department of Civil and Environmental Engineering, Carnegie Mellon University, Pittsburgh, PA 15213, USA. [12]Institut des Géosciences de l'Environnement, CNES, Grenoble, France. [13]Department of Geography & Environment, University of Lethbridge, Lethbridge, Alberta, Canada. [14]University of Silesia, Katowice, Poland. [15]Universidad Politécnica de Madrid, Madrid, Spain. ✉e-mail: will.kochtitzky@uottawa.ca

regions in the Northern Hemisphere with complete observation-based frontal ablation estimates, but these have not been updated since 2013.

Whereas many regional-scale studies have derived regional total glacier mass changes from glaciological and geodetic methods[8–11], satellite gravimetry[12,13], and numerical modeling[14–16], no study has determined the frontal ablation rate for the entire Northern Hemisphere. This has mainly been due to a previous lack of spatially and temporally consistent glacier ice thickness, surface velocity, and terminus position observations, from either field or remote sensing surveys, which is exacerbated at high latitudes.

Here we identify every marine-terminating glacier in the Northern Hemisphere, distinct from the Greenland Ice Sheet, and use measurements or estimates of ice thickness, surface velocity, and terminus position changes, together with a climatic mass balance model, to estimate mean frontal ablation (see "Methods" for details; Fig. S1). We split the observations into two periods: 2000 to 2010 and 2010 to 2020. The former decade is based on terminus position observations from summer 2000 and 2010, and velocity observations from 2000 to 2009. The latter decade is based on terminus position observations from summer 2010 and 2020, and velocity observations from 2010 to 2019. Because directly measuring frontal ablation components (i.e., calving, sublimation, and melting) is difficult, we estimate frontal ablation indirectly, by calculating ice discharge (i.e., mass flux through a flux gate perpendicular to glacier flow upstream of the terminus) and mass change due to retreat or advance (henceforth, terminus mass change), while accounting for the climatic mass balance (i.e., the mass changes due to snow accumulation, surface melt and refreezing) below the flux gate. Our computation of frontal ablation excludes submarine melting below floating tongues, but this is likely negligible as virtually no floating glacier tongues remain in the Arctic[17,18]. The residual examples are primarily within Franz Josef Land, Russia[17].

## Results and discussion
### Complete hemispheric estimates of discharge and terminus change
Guided by the Randolph Glacier Inventory version 6.0 (RGI6)[10,19], we identified 1496 marine-terminating glaciers and manually mapped their terminus positions in summer 2000, 2010, and 2020 from satellite imagery (see "Methods")[20]. The Greenland Periphery has the most marine-terminating glaciers (537), followed by the Russian Arctic (412), Arctic Canada North (252), Svalbard and Jan Mayen (160 + 6 = 166; hereafter Svalbard), Arctic Canada South (86), Alaska (42), and Iceland (1). However, the length of terminus flux gates in Russia is more than two times that of Greenland Periphery glaciers

(Table 1). North Asia has one small marine-terminating glacier on one of the De Long Islands, and we therefore include it in the Russian Arctic RGI6 region. Between 2000 and 2010, 49 of the marine-terminating glaciers lost contact with the ocean and became land-terminating with another 120 glaciers following in the next decade.

The frontal ablation rate from all 1496 glaciers was $44.47 \pm 6.23$ Gt a$^{-1}$ (all uncertainties given at the 95% confidence level, i.e., two standard deviations) between 2000 and 2010, and $51.98 \pm 4.62$ Gt a$^{-1}$ between 2010 and 2020 (Table 1). While this suggests a slight increase in frontal ablation between the two decades, the estimates are within the uncertainties of each other. Ice discharge accounted for 80 and 78% of the frontal ablation for 2000–2010 and 2010–2020, respectively, with the remaining losses due to terminus retreat. Rates of frontal ablation, ice discharge, and terminus mass loss did not vary significantly (i.e., non-overlapping uncertainties) between 2000–2010 and 2010–2020, with few exceptions (Table 1). In Alaska, terminus mass loss increased nearly tenfold while ice discharge decreased by about 15%. In Svalbard, ice discharge nearly tripled, primarily due to the surge of Basin-3 of Austfonna Ice Cap. Owing to improved data quality during 2010–2020 and smaller uncertainties, we focus primarily on estimates of this decade, unless otherwise specified.

Among the seven glacierized regions studied, the Russian Arctic experienced the highest frontal ablation rate, followed by Svalbard and Alaska (Table 1; Fig. 1). Greenland Periphery and Arctic Canada North had similar frontal ablation rates, while Iceland and Arctic Canada South were markedly lower (Table 1; Fig. 1). From 2000–2010 Hubbard Glacier in Alaska showed the highest frontal ablation rate ($4.13 \pm 0.05$ Gt a$^{-1}$) for a single glacier, but was surpassed by Basin-3 of Austfonna in Svalbard ($6.10 \pm 0.24$ Gt a$^{-1}$) during the next decade (Fig. 2).

Even though Arctic Canada North has 40% of the total Northern Hemisphere area of marine-terminating glaciers, Alaska, Svalbard, and the Russian Arctic each produce more frontal ablation. The Russian Arctic contains 22% of the marine-terminating glacier area, but it accounts for 33% of total frontal ablation.

### Uneven distribution of frontal ablation across all regions
Within each region, a few large glaciers have a disproportionate impact on the total frontal ablation over the study period (Fig. 2). In Svalbard and the Russian Arctic, 27 and 32% of glaciers (i.e., 61 and 80% of regional marine-terminating glacier area), respectively, make up 90% of the regional frontal ablation rate. A similar, small number of glaciers make up about 90% of frontal ablation in other regions as well, including 24% of glaciers in Arctic Canada North, 14% in Alaska, 28% in Greenland Periphery, and 28% in Arctic Canada South. In total, merely

## Table 1 | Frontal ablation and components

| | Frontal ablation (Gt a$^{-1}$) | | Ice discharge (Gt a$^{-1}$) | | Terminus mass loss (Gt a$^{-1}$) | | Length of flux gates (km) | Number of glaciers |
|---|---|---|---|---|---|---|---|---|
| | 2000–2010 | 2010–2020 | 2000–2010 | 2010–2020 | 2000–2010 | 2010–2020 | | |
| Alaska | $11.59 \pm 0.39$ | $10.68 \pm 0.33$ | $11.49 \pm 0.35$ | $9.79 \pm 0.18$ | $-0.1 \pm 0.17$ | $-0.89 \pm 0.28$ | 80 | 42 |
| Arctic Canada North | $4.14 \pm 1.11$ | $4.28 \pm 1.18$ | $2.68 \pm 0.65$ | $2.24 \pm 0.33$ | $-1.45 \pm 0.91$ | $-2.03 \pm 1.14$ | 481 | 252 |
| Arctic Canada South | $0.09 \pm 0.08$ | $0.09 \pm 0.08$ | $0.04 \pm 0.03$ | $0.03 \pm 0.02$ | $-0.05 \pm 0.07$ | $-0.06 \pm 0.07$ | 40 | 86 |
| Greenland periphery | $4.31 \pm 1.57$ | $3.18 \pm 1.09$ | $2.25 \pm 0.77$ | $1.88 \pm 0.43$ | $-2.05 \pm 1.36$ | $-1.29 \pm 1.0$ | 697 | 537 |
| Iceland | $0.10 \pm 0.10$ | $0.03 \pm 0.03$ | $0.10 \pm 0.10$ | $0.001 \pm 0.07$ | $-0.003 \pm 0.02$ | $-0.03 \pm 0.05$ | 5 | 1 |
| Svalbard and Jan Mayen | $7.62 \pm 2.65$ | $16.82 \pm 2.48$ | $4.88 \pm 1.98$ | $14.41 \pm 1.05$ | $-2.74 \pm 1.77$ | $-2.4 \pm 2.25$ | 657 | 166 |
| Russia: Franz Josef Land | $10.46 \pm 5.05$ | $7.44 \pm 3.29$ | $8.78 \pm 4.73$ | $4.68 \pm 2.21$ | $-1.68 \pm 1.76$ | $-2.76 \pm 2.44$ | 1377 | 328 |
| Russia: Novaya Zemlya | $2.67 \pm 1.02$ | $4.15 \pm 0.93$ | $2.02 \pm 0.84$ | $3.14 \pm 0.37$ | $-0.65 \pm 0.57$ | $-1.0 \pm 0.86$ | 182 | 39 |
| Russia: Severnaya Zemlya | $3.50 \pm 1.14$ | $5.33 \pm 0.88$ | $3.15 \pm 1.06$ | $4.23 \pm 0.59$ | $-0.36 \pm 0.43$ | $-1.1 \pm 0.64$ | 283 | 45 |
| Total | $44.47 \pm 6.23$ | $51.98 \pm 4.62$ | $35.40 \pm 5.41$ | $40.41 \pm 2.61$ | $-9.07 \pm 3.07$ | $-11.57 \pm 3.81$ | 3802 | 1496 |

Decadal mean mass losses (± uncertainties) by region (from west to east) for all Northern Hemisphere marine-terminating glaciers, due to advance or retreat of terminus position (terminus mass change), ice flow through a terminus flux gate (ice discharge), and the sum of these (frontal ablation).

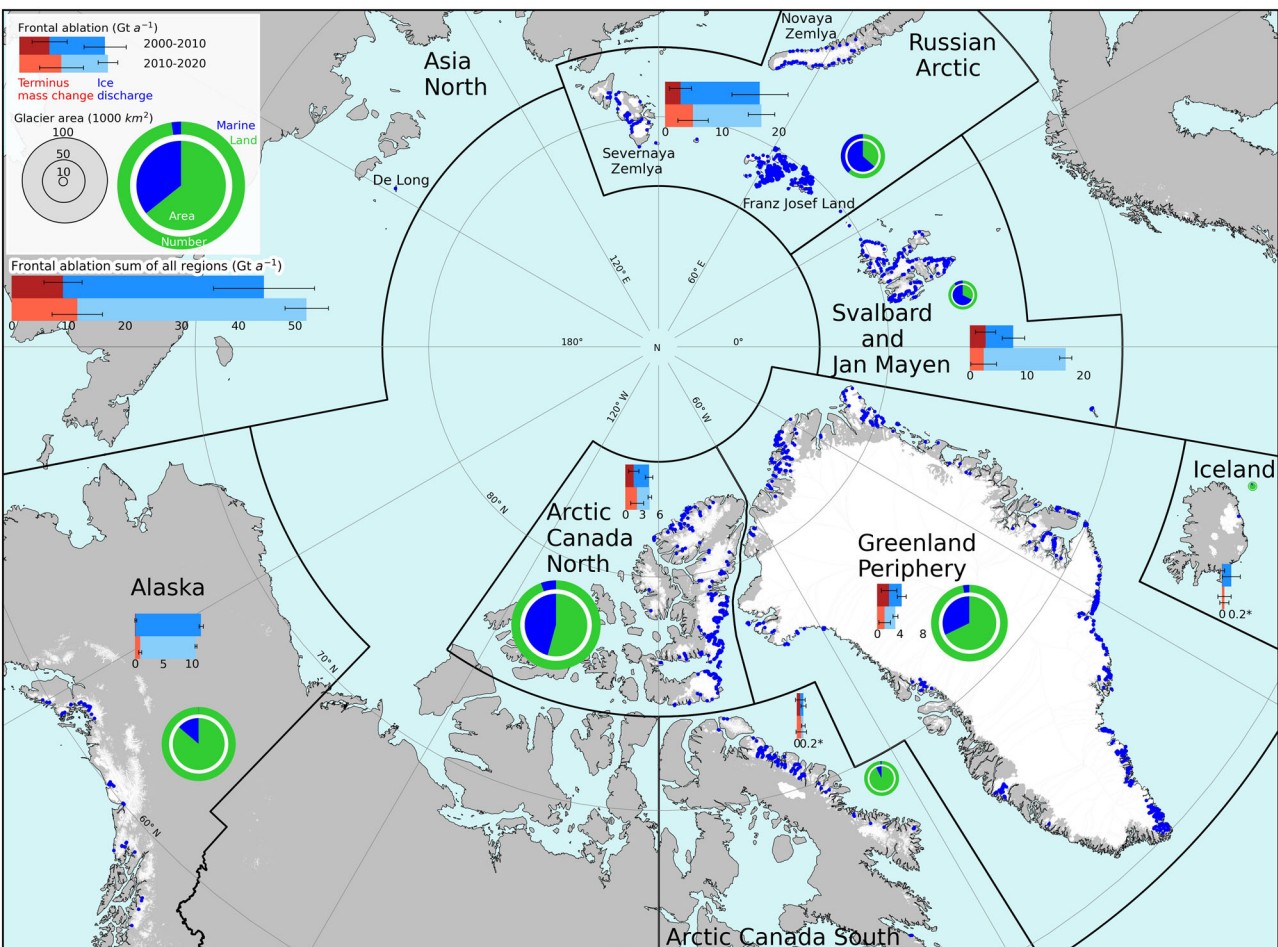

**Fig. 1 | Regional frontal ablation of marine-terminating glaciers.** Bar charts show the decadal-mean frontal ablation rates by Northern Hemisphere region from 2000 to 2020 and are divided into ice discharge and terminus mass change (uncertainties are black lines). Pie charts show the proportion of marine vs. land-terminating glaciers for each region. The Northern Hemisphere glacier proportions are shown in the legend.

1% (15 out of 1496) of marine-terminating glaciers account for 45% of Northern Hemisphere frontal ablation, and 2% (30 glaciers) account for 55% of frontal ablation.

Most marine-terminating glaciers (87% of the 1496 glaciers) contribute <0.04 Gt a$^{-1}$ to the ocean, accounting for ~14% of Northern Hemisphere frontal ablation. Hence, further studies should focus on the ~200 marine-terminating glaciers contributing more than 0.04 Gt a$^{-1}$ (see Supplemental Dataset 1), although future glacier changes (e.g., due to surging[21]; tidewater glacier cycle[22]) will modify this list.

To investigate which parts of the ocean are most affected by frontal ablation, and thus susceptible to related iceberg hazards and marine ecosystem impacts, we introduce an 'intensity index' that spatially aggregates frontal ablation from all glaciers occurring within 50 km of any point in the ocean on a 10 km grid (Fig. 3). The index depends on both the frontal ablation rates and the number of glaciers in the vicinity of each point, and thus varies in a pattern that is largely consistent with the spatial variation in frontal ablation of glaciers in each ocean region (Fig. 2). Several coastal regions show exceptionally high-intensity indices indicating hotspots of potential iceberg occurrence.

The highest intensity index occurs around northeastern Svalbard, due to the vigorous surge of Basin-3 of Austfonna which started in 2012[23], in addition to substantial contributions from the rest of the ~1200 km$^2$ ice cap. High indices are also described in the strait between October Revolution and Komsomolets islands in Severnaya Zemlya (Fig. 3), partly due to recent ice shelf break up[17,24]. In Alaska,

high indices close to Hubbard and Columbia glaciers are due to exceptionally high frontal ablation rates of these glaciers; these alone accounted for 63% of regional frontal ablation and are thus the conduits for most ice entering the Gulf of Alaska. Higher intensity indices along the western coast than the eastern coast of Novaya Zemlya is also consistent with a concentration of glaciers with high frontal ablation rates. Despite the relative low frontal ablation rates (mean 0.02 Gt a$^{-1}$) of glaciers in Franz Josef Land, the large number of glaciers yields a high index.

## Role of ice discharge versus terminus retreat

Quantifying the relative importance of ice discharge versus terminus retreat is key to understanding mass loss mechanisms and highlights where future efforts should be focused to reduce observational uncertainties. Terminus retreat was most important in Arctic Canada South, Arctic Canada North, and Greenland Periphery, where it made up 67%, 48%, and 41%, respectively, of total frontal ablation, while only amounting to 8% in Alaska. Thus, ice discharge is dominant in controlling frontal ablation rates in Alaska.

For a retreating glacier, frontal ablation is larger than ice discharge across the flux gate; conversely, for an advancing glacier, frontal ablation is lower than ice discharge. Two examples in Severnaya Zemlya illustrate the connection between terminus change and frontal ablation. We found that the eight glaciers that fed the now collapsed Matusevich Ice Shelf[24] had a frontal ablation rate of 0.20 ± 0.17 Gt a$^{-1}$ from 2000 to 2010, compared to 1.88 ± 0.23 Gt a$^{-1}$ in 2010−2020 due to the 18.91 ± 0.28 km$^2$ a$^{-1}$ terminus retreat during that decade,

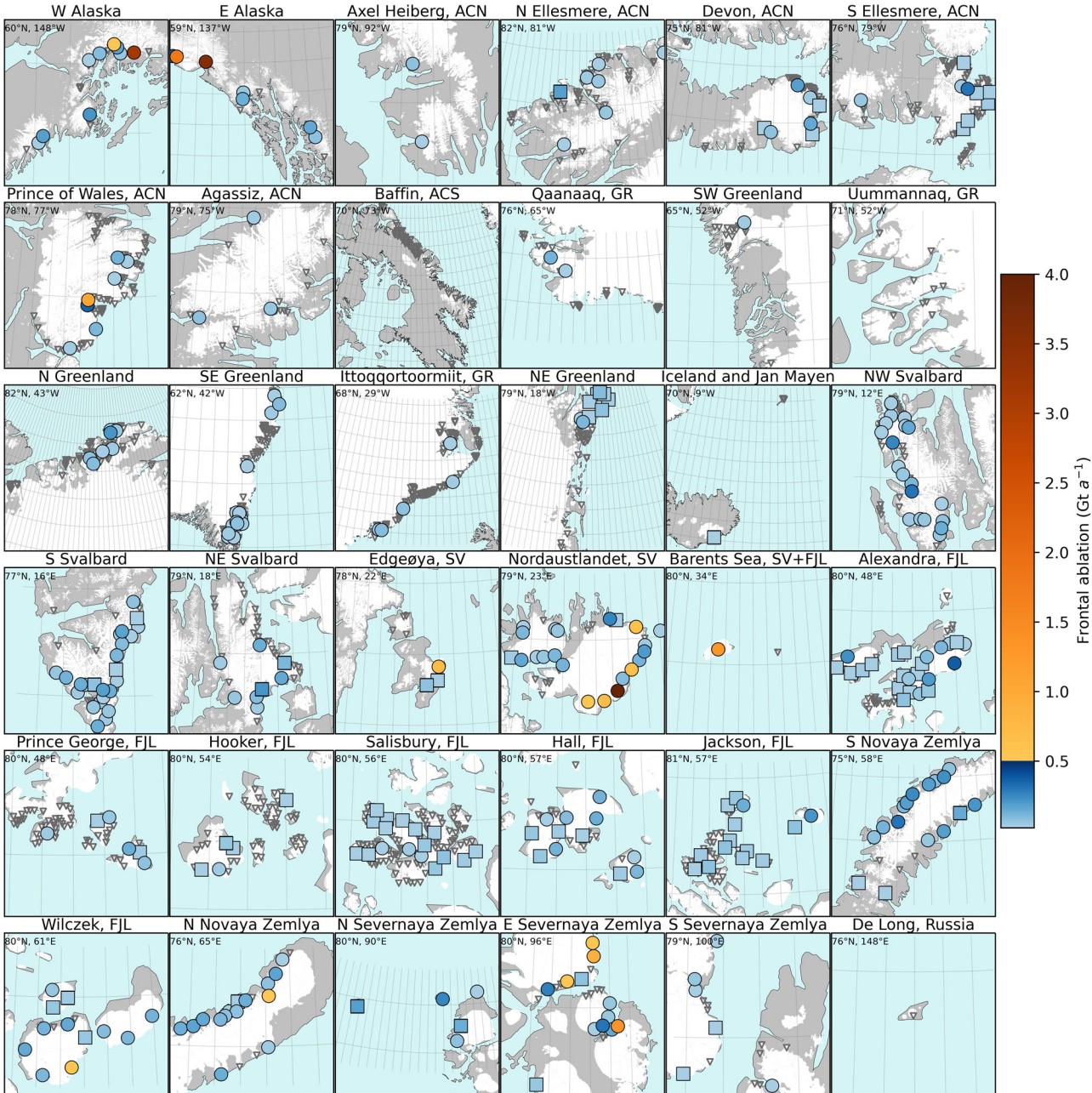

**Fig. 2 | Mean frontal ablation rate by glacier for 2010–2020.** Dark gray triangles indicate glaciers that have a frontal ablation <0.02 Gt a$^{-1}$ (not shown in color bar) and squares indicate glaciers where the uncertainty is more than 50% of total frontal ablation. Glacierized area is marked in white. Latitude and longitude in top left of each panel indicate the center point of that panel and the grid is 1° by 1° for all panels, with 1 degree of latitude equal to 111 km. The only glacier off the color bar scale is Basin-3 of Austfonna (6.18 Gt a$^{-1}$; Nordaustlandet). For uncertainties see Fig. S3. Frontal ablation for 2000–2010 is shown in Fig. S2. ACN is Arctic Canada North, ACS is Arctic Canada South, GR is Greenland Periphery, SV is Svalbard, FJL is Franz Josef Land, NZ is Novaya Zemlya, SZ is Severnaya Zemlya; other abbreviations are cardinal directions.

the largest of any Northern Hemisphere glacier. On the other hand, the surging outlet of Vavilov Ice Cap[25] discharged 1.24 ± 0.01 Gt a$^{-1}$, but frontal ablation was only 0.29 ± 0.10 Gt a$^{-1}$, due to terminus advance. Both these events are unique because the Matusevich Ice Shelf is now gone, and Vavilov Ice Cap is unlikely to surge again in the coming decades as no previous surge events have been identified since at least the 1980s, indicating a very long quiescent phase[25,26]. For the entire Northern Hemisphere, the total terminus mass loss of 11.57 ± 3.81 Gt a$^{-1}$ during 2010–2020 consisted of 2.98 ± 0.70 Gt a$^{-1}$ of terminus advance and 14.54 ± 5.97 Gt a$^{-1}$ of terminus retreat.

We examined glaciers with advancing termini for morphological indicators[27] consistent with dynamic instabilities such as surging

or pulsing (hereafter surge-type glaciers). During the surge phase, these glaciers have elevated velocities, typically by an order of magnitude or more above normal, for months to years, which can lead to increased frontal ablation over a similar period[28–30]. Of the 45 glaciers that advanced during the 2000–2010 period, we found 13 confirmed, two probable, and five possible surge-type glaciers, whereas 25 showed no signs of instability, in total accounting for 6.74 ± 0.40 Gt a$^{-1}$ of frontal ablation. From 2010–2020, 39 glaciers advanced, of which 14 were confirmed, five probable, and four possible surge-type glaciers, while 16 showed no signs of instability, in total accounting for 11.19 ± 0.71 Gt a$^{-1}$. Only 13 glaciers, six of which are surge-type, advanced over both 2000–2010 and

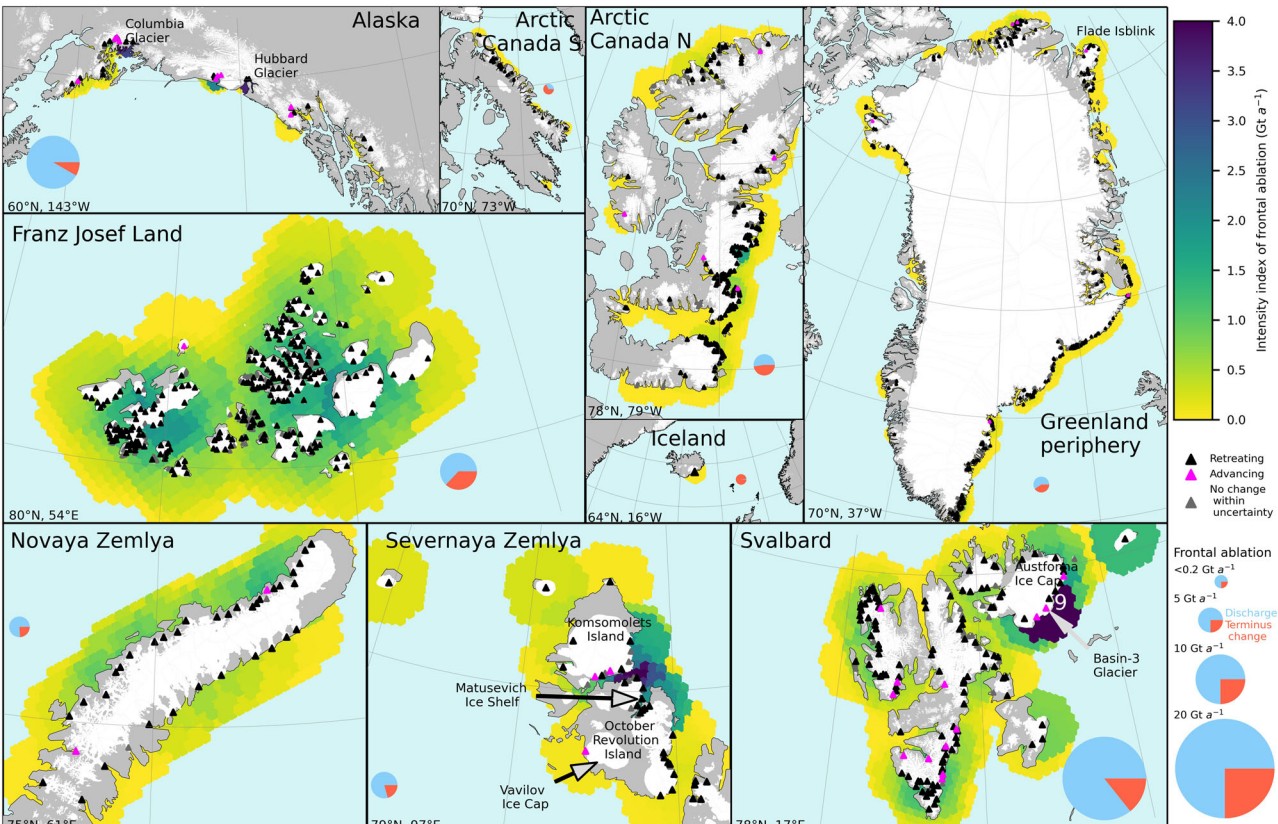

**Fig. 3 | Intensity index of frontal ablation in near-coastal ocean environments 2010–2020.** The index is defined as the sum of the frontal ablation rates of all glaciers within 50 km of their terminus, based on a 10 km ocean grid. Text in white indicates values outside the color scale. The pie charts show the proportion of ice discharge (blue) and terminus change (red) to total frontal ablation for each region, and are scaled to frontal ablation for that region. Glacier-covered area is shown in white, ocean in light blue, and other land surfaces in gray. Individual glaciers or ice caps mentioned in the text are named. The grid shows lines of latitude every 10° and longitude every 5°, or 1106 km between lines of latitude for all panels. The coordinates in the bottom left of each box show the center latitude and longitude.

2010–2020, compared to 745 glaciers that retreated during both time periods.

## Comparisons with regional mass balance

Total glacier mass balance is equal to the climatic-basal balance (the mass changes at the surface, inside, and under a glacier) and frontal ablation[1]. We subtract our frontal ablation estimates from total mass balance estimates[9] derived from elevation change observations of all glaciers in a region to estimate the regional climatic-basal balance. Since these estimates[9] do not include any mass changes below sea level due to retreat or advance, we correct these geodetic balances assuming two end-member scenarios (see "Methods") of 5.16 to 9.29 Gt a$^{-1}$ of ice lost below sea level from 2000 to 2020. This suggests that the Northern Hemisphere glacier net mass loss of 223.7 ± 14.6 Gt a$^{-1}$ from 2000–2020 by Hugonnet et al.[9] is currently underestimated by 2–4%, although this portion will not contribute to sea-level rise. We find a total mass loss due to ice discharge equivalent to 2.10 ± 0.22 mm of sea level rise between 2000 and 2020 (accounting for the effect of displacement of ocean water by changes in submarine ice, see "Methods"), with an additional 0.11 ± 0.11 mm coming from terminus mass loss above sea level.

In the Russian Arctic, frontal ablation is greater (by 6.38 ± 5.27 Gt a$^{-1}$) than the total net mass loss rate (Supplemental Table 1, Fig. S5), suggesting a positive regional climatic-basal balance. We estimate a climatic-basal balance close to zero (+1.71 ± 3.45 Gt a$^{-1}$) in Svalbard, which has overlapping uncertainties with published estimates[31,32]. All other regions have negative climatic-basal balances. Estimates of climatic-basal balance exist for one other region, Greenland Periphery[16], with which we find agreement.

While it should be possible to compare our results with recent observations of Greenland Ice Sheet ice discharge[33,34], the ice sheet currently lacks discharge estimates for approximately 77 to 211 of the slower-flowing ice sheet outlet glaciers, dependent on how outlet margins are combined or separated. Furthermore, recent studies of Greenland Ice Sheet discharge[33,34] include some peripheral glaciers and do not account for mass changes due to terminus advance or retreat, making the comparison less useful. Even so, our hemispheric ice discharge estimate is an order of magnitude lower than the most recent Greenland Ice Sheet discharge estimates[33,34].

Even though the 1496 marine-terminating glaciers comprise only 0.007% of Northern Hemisphere glaciers by count, our first Northern Hemisphere wide frontal ablation estimate indicates that they play a critical role in global glacier mass balance through mass losses by frontal ablation. Our individual glacier estimates provide valuable information for calibrating global glacier models, as well as for coastal communities affected by icebergs and related navigational hazards, and for assessing glacier impacts on near-coastal marine ecosystems.

## Methods

### Marine-terminating glacier inventory

We identified every glacier that had an interface with the ocean during at least part of a daily tidal cycle by manual examination of Landsat 5, Landsat 7, and ASTER imagery from 1999 to 2020. The only glacier not touching the ocean between 1999 and 2002 was Good Friday Glacier (Arctic Canada North), which advanced to become marine-terminating between 2000 and 2010[35], yielding a total of 1496 marine-terminating glaciers in the Northern Hemisphere.

For Greenland Periphery, beyond the ice-sheet margin, we included glaciers that were defined by Rastner et al.[36] as having either a weak (CL1) or no (CL0) hydrological connection to the ice sheet. Our total number of glaciers and terminus outlines differs from classification of the RGI due to missing glaciers[10], poorly mapped ones, and misclassified terminus types. Poorly mapped glaciers in RGI6 include those with multiple distinct termini and glaciers that share one terminus (mostly in Svalbard).

## Flux gates

We drew 5016 km of flux gates manually, although after correcting fluxgate width to be perpendicular to flow the total length is 3802 km (Table 1). We drew fluxgates with a preference for near-terminus locations where ice thickness observations exist (Fig. S1). If no thickness observations were available, flux gates were drawn as close to the terminus as practical without entering an area that was lost (median distance from the terminus in 2010 was 630 m). Flux gates are constrained horizontally by a lightly edited (to have one terminus per glacier) version of RGI6[10]. Unconstrained ice caps that terminate in the ocean, such as those found in the Russian Arctic, have one flux gate that encircles the ice cap. Flux gates were drawn approximately perpendicular to flow direction based on the orientation of medial moraines, structural patterns of other surface-flow features, and/or of valley walls. We used the x and y component velocity observations to adjust the flux gate width such that the entire width of all flux gates were perpendicular to flow[4]. Flux gates were subdivided into ~25 m segments (hereafter flux gate segments) using a unique orthographic coordinate system for each glacier to minimize the effect of distortion from regional projections. We then used these points to extract values from the velocity and ice thickness datasets.

## Frontal ablation estimation

The velocity, ice thickness, and area change measurements were combined to derive mean frontal ablation rates for every glacier in the Northern Hemisphere over the periods 2000–2010 and 2010–2020. We computed the frontal ablation rate ($A_f$; Gt a$^{-1}$, defined as positive here) from the ice discharge rate ($\dot{D}_{ice}$, Gt a$^{-1}$, defined as positive here) across a flux-gate close to the glacier terminus and the mass change rate associated with terminus position change ($\dot{M}_{term}$; Gt a$^{-1}$; defined positive for retreat and negative for advance):

$$\dot{A}_f = \dot{D}_{ice} + \dot{M}_{term} \qquad (1)$$

$\dot{D}_{ice}$ is given by:

$$\dot{D}_{ice} = \left( \rho \left( \sum_{n=1}^{N} (V_n \cdot H_n \cdot d_n) \right) - (S_f \cdot \dot{B}_{clim}) \right) \qquad (2)$$

where $\rho$ is the vertically averaged density of the ice column (900 kg m$^{-3}$)[37], $N$ is the total number of flux gate segments along the flux gate, $V_n$ is the vertically averaged ice velocity component normal to the flux gate at segment $n$ (assumed to be 95% of the surface velocity at that location; m a$^{-1}$)[37], $H_n$ is the ice thickness (m), $d_n$ is the flux gate segment width, $S_f$ is the area of the region below the flux gate not involved in terminus retreat or advance (m$^2$; area shown in Fig. S1A), and $\dot{B}_{clim}$ is the mean specific climatic mass balance rate (kg m$^{-2}$ a$^{-1}$; defined as positive here).

We calculated the mass change due to terminus advance or retreat by:

$$\dot{M}_{term} = \left( \rho \cdot \frac{\triangle S_{term}}{\triangle t} \cdot \bar{H} + \triangle S_{term}/2 \cdot \dot{B}_{clim} \right) \qquad (3)$$

where $\bar{H}$ is the mean thickness of the terminus in the area gained or lost (in m) and $\triangle S_{term}$ is the area (m$^2$) of the glacier that is now replaced by

the ocean for retreating glaciers, or which occupies previous ocean area for advancing glaciers (division by 2 is necessary to account for the average area over the decade as no area was lost at the start of decade), and is computed from the digitized termini positions in 2000, 2010, and 2020 (Fig. S1). $\triangle t$ is the number of years between observations.

## Ice surface velocity

Ice surface velocities were primarily derived from Landsat-based, annual displacement mosaics of the Inter-mission Time Series of Land Ice Velocity and Elevation (ITS_LIVE; Fig. S1C)[38] at 240 m resolution[39]. Data gaps smaller than one pixel were filled from the nearest neighbor. To improve the Greenland Periphery and Russian Arctic observations, we included MEaSUREs InSAR[40] and Sentinel−1 data[41], respectively, when the signal-to-noise ratio was lower than that of ITS_LIVE. For glaciers with no ITS_LIVE observations, such as those above 82.7°N, we used Landsat or Sentinel-2 imagery from 2016–2020 and AutoRIFT[42] to produce velocity maps. For a small ice cap in the Barents Sea (300 km$^2$) and a glacier in the De Long Islands (10 km$^2$), we could not generate reliable velocity results, and thus assumed a 5 ± 5 m a$^{-1}$ velocity based on evidence of little ice motion (i.e., no flow bands). For Svalbard and the Russian Arctic, where the optical velocity record from 2000 to 2010 was sparse in ITS_LIVE, we filled this gap using 1992–2012 observations from the JERS1, ERS1, ALOS PALSAR, and TerraSAR-X satellites. We used synthetic-aperture radar (SAR) offset-tracking to compute winter velocity from these scenes at 100 m resolution[43]. We combined all available velocity observations over each decade to derive a mean decadal velocity along the flux gate for each glacier. We do not account for seasonal variability in ice motion due to a lack of data to quantify this variability. In the few instances for which no velocity data was available for the 2000–2010 period (3.9% of flux gates), mean decadal values from 2010 to 2020 were used to fill the gaps.

## Ice thickness

Direct radar-derived measurements, primarily from the Glacier Thickness Database (GlaThiDa) 3.0.3[44], were used to determine ice thickness along the flux gates and over deglaciated areas due to terminus retreat since 2000. For Arctic Canada North, Center for Remote Sensing of Ice Sheets[45] radar tomography observations acquired in ~3 km swaths from the 2014 NASA Operation IceBridge mission provided the main ice thickness source. Additional radio-echo sounding observations were incorporated for the Russian Arctic, where available[46–52]. Whereas most of our observations were acquired during the study period, we used 1288 Svalbard thickness measurements in GlaThiDa from 1980 to 1995 and some observations in Russia that were collected in 1994[46] and 1997[47]. To ensure thickness data are contemporaneous, we used elevation change estimates from Hugonnet et al.[9] to correct ice thickness at the fluxgate from the time of the measurement or model to 2005 and 2015 for use in ice discharge calculations. Where point thickness observations exist, we averaged all observations within 100 m of each flux gate segment to derive the thickness at that point (Eq. 2). When only a centerline observation was available, or the thickness observations did not cover the entire length of the flux gate, we assumed a U-shaped valley to determine thickness where no observations are available[53]:

$$H_n = \frac{10 - H_{center}}{W_1} * W_2{}^2 + H_{center} \qquad (4)$$

where $W_1$ (m) is distance from the glacier margin and $W_2$ (m) is distance from the glacier centerline with thickness $H_{center}$ (m). We use 10 m as a minimum ice thickness on the glacier margins. Similar approaches can be found in Sánchez-Gámez and Navarro[54]. This may not accurately reflect the glacier thickness if the centerline observation is not the

deepest point on the glacier, or if the glacier valley is not U-shaped, although we have no way to quantify this.

In all, there is at least one ice thickness observation for 268 glaciers, which together contribute 69% of the total frontal ablation from 2010 to 2020. Out of these 268 glaciers, 250 have at least a centerline observation (middle 20% of glacier), 161 have at least 75% coverage across the flux gate, and 120 have at least 90% coverage across the flux gate. When no thickness observations were available, data gaps were filled with modeling results from Millan et al.[55] (Fig. S1B). We found that the model results overestimated glacier thickness at the fluxgate, even after accounting for elevation changes with an average 135 m bias between model estimates and observations. We empirically removed this bias by modeling it as linear function of the glacier thickness to match existing measurements.

For five small glaciers where no modeled thickness data were available, we assigned a similar thickness to that of a morphologically similar nearby glacier. For small glaciers, modeling occasionally produces ice thickness values that are unrealistically small (e.g., <5 m average). When the model results suggested that the average ice thickness along the flux gate is <30 m, we replaced these values with 30 m and assigned an uncertainty of 20 m (for 144 glaciers during 2000–2010 and 230 glaciers during 2010–2020). For the six marine-terminating glaciers on Jan Mayen we used scarce point measurements[55] and flow-law theory[37] to estimate the thickness.

To determine mean ice thickness $\bar{H}$ (Eq. 3) over the glacier area that was lost or gained we averaged all available observations. Where no observations were available, we used modeled thickness estimates (debiased in the same way as above) for the terminus area that was gained or lost unless these data were not available, in which case we assumed the thickness to be 60 ± 30% of the mean thickness along the flux gate. On 74 glaciers, primarily in East Greenland, model thickness estimates were unrealistically low (<10 m) in an area that was gained or lost, so we replaced those values with 60 ± 30% of the mean thickness along the flux gate.

## Glacier terminus front positions

To determine $\triangle S_{term}$ (Eq. 3) for the periods 2000–2010 and 2010–2020, glacier terminus positions were digitized manually for every glacier in approximately 2000, 2010, and 2020, primarily from summer, cloud-free, true-color Landsat 5, 7, and 8 imagery (30 m spatial resolution). When Landsat images were not available, we first attempted to fill the gap with true-color ASTER data (15 m spatial resolution), particularly in northern Greenland and Arctic Canada North for 2000 and 2010. Where no cloud-free optical observations were available, we used Radarsat-1 fine beam mode (2000; 8 m pixel resolution; used in Northern Greenland) and ALOS PALSAR fine beam single polarization (2010; 10 m pixel resolution; used in Arctic Canada North) imagery downloaded from the Alaska Satellite Facility. Mapping was carried out in QGIS version 3.10 in a WGS84 Arctic Polar Stereographic projection (EPSG 3995). To measure the area of each polygon we reprojected the dataset into a unique orthographic projection centered on each glacier to eliminate the impact of distortion.

## Climatic mass balance below the flux gate

To account for mass changes unrelated to frontal ablation (e.g., surface melting) between the flux gate and glacier front), we used the Python Glacier Evolution Model (PyGEM)[56] to model the climatic mass balance $\dot{B}_{clim}$ for each glacier. $\dot{B}_{clim}$ is defined as the mass changes due to snow accumulation, and melt at the surface and refreezing of melt and rain water below the surface[1]. PyGEM assumes the basal mass balance and internal ablation is negligible. We calibrated PyGEM using geodetic mass balance estimates[9]. We used monthly ERA5 air temperature and precipitation data[57] to compute glacier melt with a degree-day model, accumulation with a temperature threshold, and refreezing from annual air temperature. We extracted $\dot{B}_{clim}$ for the

lowest two 10-m elevation bins of each glacier. The glacier area was assumed constant in PyGEM. For two well-investigated glaciers with high frontal ablation rates we used estimates of climatic balance from direct observations: Columbia Glacier, Alaska (8 ± 2 m water equivalent)[58], and for Basins 2 and 3 of Austfonna, Svalbard (−0.6 ± 0.3 m water equivalent)[32,59]. For glaciers lacking appropriate input data (7 small glaciers), we used regional (defined by RGI primary regions) averages of the climatic mass balance.

For 490 glaciers from 2000 to 2010 and 602 glaciers from 2010 to 2020, the absolute $\dot{B}_{clim}$ was greater than the ice discharge, which is physically impossible. For 369 (2000–2010) and 406 (2010–2020) of these glaciers, we increased the average discharge and reduced the average climatic balance within uncertainties to constrain each variable to real-world possibilities. For the remaining glaciers, we used ice discharge values neglecting $\dot{B}_{clim}$, but with 100% uncertainty. These 488 and 601 glaciers only made up 0.70 ± 0.44 Gt a$^{-1}$ (2000–2010) and 0.54 ± 0.46 Gt a$^{-1}$ (2010–2020) of ice discharge, accounting for 2% of total ice discharge during both time periods. See Dataset S1 for a list of glaciers that have this correction.

## Surge-type glaciers and instabilities

We individually examined glaciers that exhibited a terminus advance from 2000–2010 and/or 2010–2020 to determine if the advance was due to a dynamic instability such as pulsing or surging. We classified the glaciers according to the RGI classification scale (0: no evidence, 1: possible, 2: probable, 3: observed) based on glacier morphology (including tear-drop moraines, pothole fields, crevasse patterns)[27] viewed from satellite imagery and literature where available[26,60-64].

For the Nathorstbreen system and an outlet of the Leningradskiy Ice Cap, our measured discharge was greater than terminus mass gain. However, we adjusted these totals within their uncertainties to make physical sense. For six glaciers from 2000 to 2010 and one from 2010 to 2020, the discharge was less than the terminus change; however, in each instance the mass held in terminus advance was <0.02 Gt a$^{-1}$, so we artificially increased discharge to be within the uncertainty of terminus change while assigning 100% uncertainty to the discharge. These adjusted glaciers accounted for a total frontal ablation of 0.02 Gt a$^{-1}$.

## Sea-level rise calculations

We estimated the sea-level equivalent of frontal ablation assuming a global ocean area[1] of 362.5 × 10$^6$ km$^2$. To account for mass changes below sea level that do not contribute to sea-level change as the ice displaces ocean volume, we subtract the mass losses due to retreat below sea-level or add the gains due to advance. Since the ice fraction of total ice thickness below sea level is unknown, we calculate a higher bound scenario (assuming a fraction of 90% which corresponds to ice close to flotation) and a best estimate lower-bound scenario (50%). Estimates of mass loss below sea level are typically excluded from current global mass balance estimates[9]. Our frontal ablation estimates alone are not able to provide a total sea level rise estimate because frontal ablation is only one component of mass balance.

## Uncertainty in frontal ablation estimates

We estimate the uncertainty in ice discharge $\sigma_{\dot{D}}$ assuming each uncertainty component (Eq. 2) is independent, which yields:

$$\sigma_{\dot{D}}^2 = \rho_{ice} \sum_n \left( \sqrt{\left(\frac{\sigma_{V_n}}{V_n}\right)^2 + \left(\frac{\sigma_{H_n}}{H_n}\right)^2 + \left(\frac{\sigma_{d_n}}{d_n}\right)^2} \cdot V_n H_n d_n \right)^2 + \left(S_f \sigma_{\dot{B}_{clim}}\right)^2 + \left(\sigma_{S_f} \dot{B}_{clim}\right)^2 \tag{5}$$

where $\sigma_{V_n}$ is the uncertainty in vertically integrated velocity (assumed to be 95% of the surface velocity at that location)[37], $\sigma_{H_n}$ is the

uncertainty in ice thickness, $\sigma_{d_n}$ is the fluxgate width uncertainty, all associated to the flux gate point $n$. At the glacier-scale, $\sigma_{\dot{B}_{clim}}$ is the uncertainty in climatic specific-mass balance, and $\sigma_{S_f}$ is the uncertainty in the area below the flux gate. Similarly, we estimate the uncertainty in terminus mass change $\sigma_{\dot{M}_{term}}$ assuming each uncertainty component (Eq. 3) is independent, which yields:

$$\sigma^2_{\dot{M}_{term}} = (\sigma_{\Delta S_{term}} \underline{H})^2 + (\sigma_{\underline{H}} \Delta S_{term})^2 + \left(\sigma_{\dot{B}_{clim}} \frac{\Delta S_{term}}{2}\right)^2 + \left(\dot{B}_{clim} \frac{\sigma_{\Delta S_{term}}}{2}\right)^2 \tag{6}$$

where $\sigma_{\Delta S_{term}}$ is the uncertainty in the terminus area change and $\sigma_{\underline{H}}$ is the uncertainty in the mean thickness over $\triangle S_{term}$. Finally, the uncertainty in frontal ablation $\sigma_{\dot{A}}$ is estimated by considering the ice discharge and terminus mass change uncertainties (Eq. 1) as independent:

$$\sigma^2_{\dot{A}} = \sigma^2_{\dot{D}} + \sigma^2_{\dot{M}} \tag{7}$$

### Uncertainty sources at flux gate point or glacier-scale
We relied on the uncertainties reported by each dataset, whether at the scale of a flux gate point, flux gate, or glacier to propagate the uncertainties through Eqs. 5 and 6.

At flux gate points, we found good agreement (<50% difference) between the average velocity uncertainties $\sigma_{V_n}$ reported in ITS_LIVE and the average dispersion between the ITS_LIVE and MEaSURES estimates at the same flux gates. We found a similar agreement (<30% difference) between measured and modeled ice thicknesses[55], after accounting for elevation changes[9] and debiasing (see Supplemental Material for details). The SAR-derived velocity uncertainties average 10 m a$^{-1}$ for TerraSAR-X and ALOS PALSAR[43], 20 m a$^{-1}$ for JERS-1[65], and 40 m a$^{-1}$ for ERS[66]. We derived the uncertainty in flow direction based on the stated uncertainties in the x and y velocity products, translated into a flux gate segment width uncertainty $\sigma_{d_n}$. When these data were unavailable, we assumed a 10° uncertainty in flow direction.

Unless otherwise reported, we assumed a 10% uncertainty in ice thickness $\sigma_{H_n}$. We also assumed a 10% uncertainty for ice thicknesses derived from U-shaped valley-based modeling from a centerline depth measurement[53]. When a model depth or thickness observation in the terminus region was not available to calculate terminus mass change, we assumed that the average thickness of the area gained or lost is $60 \pm 30\%$ of the average flux gate thickness.

The uncertainty in area gained or lost $\sigma_{S_f}$ was assumed to be one pixel and derived by multiplying the length of the perimeter of the changed terminus area by the width of a pixel (30 m for Landsat and 15 m for ASTER)[67].

We estimated the uncertainty in the climatic specific-mass balance from the median absolute deviation of 50 simulations from Bayesian calibration of model parameters[68] forced with ERA5 air temperature and precipitation[57] calibrated on 2000–2019 glacier mass balances[9] for each glacier. We assumed the uncertainties in climatic mass balance are independent between glaciers.

### Spatial correlations in ice thickness and velocity uncertainties
We quantified the spatial correlation of velocity and ice thickness uncertainties (Supplemental Fig. 6) by estimating global-scale, empirical variograms[69,70] using the difference between independent sources of estimates at the same flux gates. We performed the spatial correlation analysis at distances covering several orders of magnitude (from 25 m to 1000 km), thereby accounting for biases at glacier and regional scales through long-range correlations[9].

We compared ITS_LIVE and MEaSUREs velocity observations in Greenland (Supplemental Fig. 7) and found fully correlated (100%) variance at short distances (<50 m), and correlated variances at 40–100% within ~700 m. We attribute these short-range correlations to

the resolution of optical imagery and image-matching. Velocity estimates remain correlated at 15–40% within 25 km and 0–15% within 1000 km, highlighting moderate glacier and regional-scale errors due to seasonal differences when estimating yearly velocity. Beyond 1000 km, the velocity estimates are fully decorrelated.

We compared all available ice thickness measurements to debiased model estimates[55] (Supplemental Fig. 8). Ice thickness variance was correlated at 80–100% within 2 km, the result of short-scale modeling artefacts. Ice thickness variance remains correlated between 50 and 80% within 150 km, and between 0 and 50% within 1000 km, implying large-scale biases in modeled ice thicknesses that are likely owed to the temporal inconsistency of glacier outlines. After 1000 km, ice thickness estimates are fully decorrelated.

More details on spatial correlations are available in the Supplementary Material.

### Uncertainty propagation from pixel to glacier and regional scales
For the ice discharge uncertainty (Eq. 5), we independently considered the ice discharge at the flux gate $\dot{D}_{gate}$ based on velocity and ice thickness estimates (Eq. 1):

$$\dot{D}_{gate} = \sum_n \dot{D}_n = \sum_n \rho_{ice} V_n H_n d_n \tag{8}$$

We propagated the uncertainties in ice discharge at the flux gate from flux gates $n$ to glaciers, and then repeated the same approach from glacier to region, accounting for spatial correlations in velocity and ice thickness uncertainties[9,71]:

$$\sigma^2_{\dot{D}_{gate}} = \sum_n^N \sigma^2_{\dot{D}_n} + \sum_n^N \sum_{n \neq m}^N C_{\dot{D}}(l_{n-m}) \sigma_{\dot{D}_n} \sigma_{\dot{D}_m} \tag{9}$$

where $C_{\dot{D}}(l)$ is the spatial correlation in ice discharge at the spatial lag $l$, and $l_{n-m}$ is the distance between flux gates $n$ and $m$ in the glacier, or between glacier $n$ and $m$ in the region. The correlation in ice discharge $C_{\dot{D}}(l)$, ranging from 0 to 1, is estimated either for each glacier or region based on the velocity and ice thickness empirical variograms (Supplemental Figs. 7 and 8).

For the terminus mass change uncertainty, we applied a similar approach to propagate the uncertainty in average ice thickness over the area gained or lost $\sigma_{\underline{H}}$, based solely on the ice thickness variogram (Supplemental Fig. 8).

## Data availability
All relevant data outcomes from this study are available in Dataset S1. The python code developed for this project can be downloaded at github.com/willkochtitzky/FrontalAblation. Glacier terminus positions can be downloaded at the Polar Data Catalogue: polardata.ca/pdcsearch/PDCSearch.jsp?doi_id=13257.

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

## Acknowledgements

The start of this project and early ideas that informed this work were a result of the "Workshop on the Importance of Calving for the Mass Balance of Arctic Glaciers" held in Sopot, Poland in 2016, sponsored by the Centre for Polar Studies, University of Silesia and the IASC Network on Arctic Glaciology. We thank Ruitang Yang for her help in improving the error analysis. W.K. acknowledges support from the Vanier Graduate Scholarship. L.C. thanks the Natural Sciences and Engineering Research Council of Canada, University of Ottawa and ArcticNet Network of Centres of Excellence Canada for funding. T.S. acknowledges support from the ESA Glaciers CCI project. A.G. and I.L. thank RG State Contract FMGE-2019-0004, and H.J. the Natural Sciences and Engineering Research Council of Canada. Re.Ho. and D.R.R. acknowledge support from the NASA grant 80NSSC20K1296.

## Author contributions

W.K. did most of the data collection, performed the analyses and calculations, and wrote the initial manuscript draft. L.C. aided in data collection and helped write the initial manuscript. W.W., J.D., A.C., A.D., H.J., J.J., and F.N. helped with initial project planning and contributed to the final manuscript. Re.Ho. contributed substantially to writing the final manuscript draft. J.D., T.B., A.G., and I.L. provided ice thickness data for the Russian Arctic. T.Z. provided velocity observations in Svalbard and the Russian Arctic. H.J. and J.C. provided frontal ablation estimates for Jan Mayen. D.R.R. provided melt model results, R.M. provided glacier thickness data, W.K., Re.Ho., and Ro.Hu. developed the error analysis and RoHu completed it.

## Competing interests

The authors declare no competing interests.
