## [Peer Review File · Nature Communications]

The unquantified mass loss of Northern Hemisphere marine-terminating glaciers from 2000-2020Reviewers' Comments:

Reviewer #1:

Remarks to the Author:

Dear Nature Editor(s) and Manuscript Authors,

In this manuscript, the authors provide a comprehensive review of the frontal ablation on 1496 marine-terminating glaciers – all such glaciers in the northern hemisphere were aren't hydrologically connected to the Greenland ice sheet. One of the main results of this study is that frontal ablation, a process independent of surface melt, has been responsible for a significant fraction of recent global sea level rise. The authors partition overall frontal ablation (including submarine melt, calving, 'subaerial melt', and sublimation) statistics between regions, and between front changes and discharge.

Overall, this study is extensive, and I would expect it to be cited often, both in scientific literature and in comprehensive assessment reports such as in the next IPCC. I have two major comments, added here, and a few minor line comments, appended below.

First, given that this paper will likely be cited often in terms of sea level budgets, many of the statistics in this paper ought to be detailed in a broader context. For example, how does the ~3.6 mm of sea level rise from these glaciers compare to that of other mass-driven changes, for example from surface melt on these glaciers? Or from similar processes on the Greenland Ice Sheet? Or from all melted ice over 2000-2020? The sea level budgets from Frederikse et al 2020 would likely be helpful in this exercise.

Second, I'm not quite convinced that Figure 3 fits with the discussion or is entirely that useful. The intensity index maps are sold in this paper for their connection to iceberg hazards or impacts on marine ecosystems (line 134). Yet, many of places where the highest intensity indices have been calculated either have sea ice in the region or are close to the Greenland ice sheet which is also adding lots of icebergs to the ocean, or both. In effect, ocean vessels will already be well aware of ice in the region, and modelers or biologists would likely just use the point-source flux values directly.

I think a more useful final figure would give some context to contributions from frontal ablation relative to other processes. For example, how do the frontal ablation values compare to "climatic-basal" mass balance (line 185)? A figure demonstrating the statistical and/or spatial distribution of this type of comparison would help us understand how to interpret the frontal changes, and the data is already accessible in this manuscript in supplementary table S1.

As mentioned above, this is an extensive study, and I look forward to seeing it in print after these comments are addressed. If desired, I would also be happy to review a second draft of this manuscript.

Sincerely,
Michael Wood

Lines 33-34: This first sentence is a little strange because 100% of the glacier and ice cap area eventually drains to the ocean, right? A suggested revision: "In the Northern Hemisphere, nearly 1500 glaciers, accounting for 28% of all glacier and ice cap area outside of the Greenland Ice Sheet, terminate directly in the ocean. At this ice-ocean interface, glaciers lose ice via melting, calving, and sublimation, yet the collective mass loss from these frontal ablation processes has not been comprehensively quantified. Here, we estimate decadal frontal ablation rates on these glaciers using measurements of ice discharge and terminus change from 2020-2022. ... "

Line 42: quantify the "remainder" too

Line 43: It would be helpful in the abstract if the ~ 3.6 mm of SLR was put into context of total global or barystatic or ice-driven SLR.

Line 81: "are remaining" -> "remain"

Line 155: It may be personal ignorance, but I haven't seen "kinematics" used often in the literature. It's implicit here that this relates to terminus retreat, but perhaps this can be made more obvious for folks like myself by rewording, such as "Frontal ablation is the combination of two processes: a dynamic component related to variations in ice discharge, and a kinematic component related to variations in the terminus position. Quantifying the partition between these mass loss mechanisms is key to understanding underlying drivers of frontal ablation and highlighting where future efforts should be focused to reduce observational uncertainties."

Line 169: Why is the ice cap not expected to surge again in the next few decades? Is there a citation for this?

Line 277: Please describe the division by 2 being necessary to account for average area more explicitly.

Line 280-297: How do you avoid sampling biases in the velocity measurements? E.g. some sensors provide observations in summer or winter only. Are seasonal variations in velocity enough to significantly change the results for a particular glacier?

Line 317: Where does the 10 come from?

Line 364: It would be helpful to define climatic mass balance again here (the only other time its defined is in line 185). It would also be helpful to provide a few more sentences about how the model calculated this mass balance. The surface components are easier to understand with the provided information about melt, accumulation, and refreeze, but what is the model doing to calculate this "inside" or under the glacier? If nothing, then this term should be referred to as SMB rather than climatic mass balance. If something, then info on the partitioning between these processes would shed some light on how the model is performing.

Line 376: How does Bclim from the model compare to Bclim from the observations on the well-investigated glaciers? What does this tell us about the uncertainty of the Bclim estimates from the model?

Reviewer #2:

Remarks to the Author:

What are the noteworthy results?

This paper is the first to quantify loss at the ice/ocean interface of all northern hemisphere glaciers not connected to the Greenland ice sheet. This required a huge amount of data compilation and synthesis. While the total estimate was similar to the previous estimate, the regional results constrain the spatial variability in frontal ablation, which is interesting.

Will the work be of significance to the field and related fields?

This work confirms that front ablation of northern hemisphere glaciers (outside greenland) is a relatively small component of mass balance and insignificant in terms of sea level rise. However, such confirmation is important for the field, and the estimates of discharge may be important for studies of

fjord ecology (as mentioned in the text).

How does it compare to the established literature? If the work is not original, please provide relevant references.

The work confirms a previous estimate of glacier frontal mass balance based on empirical modeling (Huss and Hock, 2015, cited in text). This paper estimates change between 2000-2010 and 2010-2020, finding an increase within the error.

Does the work support the conclusions and claims, or is additional evidence needed?

Line 37: The fact that this study only assess Northern Hemisphere glaciers should be specified in the title.

Line 40: It's worth noting that the increase in ablation between the 10-year periods (~ 4 Gt/yr) is well within the error of 10 Gt/yr), despite the large glacier retreats & thinning.

Line 213-215: The basis for the conclusion "our first Northern Hemisphere wide frontal ablation estimate indicates that they play a critical role in global glacier mass balance through mass losses by frontal ablation" needs to be more clearly demonstrated. How does the ~ 50 Gt/yr of ice loss (which is within the error of the Huss and Hock (2015) estimate) compare to surface melt? (i.e. what's the % of accumulation lost to ablation)? In terms of sea level rise, 0.15 mm/yr is within the uncertainty of current total estimates.

Are there any flaws in the data analysis, interpretation and conclusions? Do these prohibit publication or require revision?

line 153+: "Kinematics and Dynamics" It's not immediately clear what is specifically kinematics and dynamics refer to. My understanding is that kinematics and dynamics are two different ways to estimate a flux, with the former using continuity (e.g. mass balance) and the later using a description of the physical process (e.g. a flow model). Thus, either could describe advance/retreat or discharge (as far as I can tell, only kinematics are used in this paper). Please be more clear in your terminology here.

Is the methodology sound? Does the work meet the expected standards in your field?

Yes.

Is there enough detail provided in the methods for the work to be reproduced?

Yes.

We thank Dr. Wood and anonymous reviewer #2 for their helpful comments on our manuscript. We appreciate the ideas to improve this study and have incorporated nearly all their suggestions. One of the big challenges in writing this manuscript has been making comparisons to measurements of mass balance elsewhere, as both reviewers have described. We include an explanation here that describes some of that conversation. We have submitted both a clean word document without track changes and a PDF showing all the changes we have made to the manuscript. The line numbers referenced below refer to this PDF.

Reviewer #1 (Remarks to the Author):

First, given that this paper will likely be cited often in terms of sea level budgets, many of the statistics in this paper ought to be detailed in a broader context. For example, how does the ~3.6 mm of sea level rise from these glaciers compare to that of other mass-driven changes, for example from surface melt on these glaciers? Or from similar processes on the Greenland Ice Sheet? Or from all melted ice over 2000-2020? The sea level budgets from Frederikse et al 2020 would likely be helpful in this exercise.

This is a subject that we have spoken about at great length amongst the co-authors over the past year. The main problem is that comparing frontal ablation as a volume (aka sea level rise equivalency) to any other volume loss is not comparing similar terms. This problem arises because frontal ablation is only one component of total mass loss and thus accounting for which portion of accumulation should be assigned to balance out surface melt and frontal ablation is complicated if not impossible. An additional issue is that various papers compute mass balance terms in different ways (e.g., whether surface mass balance includes superimposed ice), making comparisons between their results and ours challenging. Considering this, we strictly only use mass balance terms as defined in Cogley et al. (2011), and have chosen to compare our results to the climatic-basal balance, which still provides valuable insight.

Based on the equations in Cogley et al. (2011), total mass balance is equal to either accumulation and ablation, or climatic-basal balance and frontal ablation. At present, studies of geodetic mass balance (Hugonnet et al., 2021) or direct measures of mass change (Ciraci et al., 2020) provide the best estimates of total mass balance, although these estimates both exclude mass lost below sea level (our estimates do not exclude this mass). Directly estimating accumulation and ablation as terms is complicated, and still requires knowledge of frontal ablation. It is easier, and more available in the literature, to find estimates of the climatic-basal balance for comparisons with frontal ablation and total mass balance. At present, models are the best way to estimate climatic-basal balance, but these only exist for some regions (e.g. Noel et al., 2020; Schuler et al., 2020).

By comparing frontal ablation, and total mass balance, we can understand both the sign and approximate magnitude of climatic-basal balance, which should help inform and improve modeling efforts (Figure S5). In our understanding of mass balance, this is the only truthful way to compare our estimates of frontal ablation with other estimates of mass loss and sea level rise. Thus, we only provide the conversion to sea level rise equivalency to add some context to our estimates. To make it clearer that these are just volumetric conversions, we changed “account for” to “is equivalent to” in the abstract. We have also changed similar language elsewhere in the text (line 239). This should clarify that frontal ablation is not directly responsible for this sea level rise, but is equivalent to that volume.

Second, I'm not quite convinced that Figure 3 fits with the discussion or is entirely that useful. The intensity index maps are sold in this paper for their connection to iceberg hazards or impacts on marine ecosystems (line 134). Yet, many of places where the highest intensity indices have been calculated either have sea ice in the region or are close to the Greenland ice sheet which is also adding lots of icebergs to the ocean, or both. In effect, ocean vessels will already be well aware of ice in the region, and modelers or biologists would likely just use the point-source flux values directly.

I think a more useful final figure would give some context to contributions from frontal ablation relative to other processes. For example, how do the frontal ablation values compare to "climatic-basal" mass balance (line 185)? A figure demonstrating the statistical and/or spatial distribution of this type of comparison would help us understand how to interpret the frontal changes, and the data is already accessible in this manuscript in supplementary table S1.

We also hope that modelers and others will use the point-source flux information directly. We agree that a figure showing the comparison of frontal ablation to climatic-basal balance would be incredibly informative. However, to our knowledge individual glacier estimates of climate-basal balance do not exist and are only available for some regions, making this comparison impossible. We hope to be able to do this with future work in collaboration with many others.

Figure 3 provides estimates of frontal ablation spatially and from the ocean perspective, and we believe that it is useful to show patterns that are not obvious in Figs. 1 or 2, such as where there are many glaciers in a small area that contribute a relatively small amount of frontal ablation each that when summed together are quite significant (e.g., as in Franz Josef Land or parts of Arctic Canada). Figure S5 shows what is possible in respect to a comparison between total mass balance and frontal ablation, but we think that Figure 3 better illustrates wide variability in frontal ablation across the hemisphere. For these reasons, our preference is to maintain the current figure 3.

Lines 33-34: This first sentence is a little strange because 100% of the glacier and ice cap area eventually drains to the ocean, right? A suggested revision: "In the Northern Hemisphere, nearly 1500 glaciers, accounting for 28% of all glacier and ice cap area outside of the Greenland Ice Sheet, terminate directly in the ocean. At this ice-ocean interface, glaciers lose ice via melting, calving, and sublimation, yet the collective mass loss from these frontal ablation processes has not been comprehensively quantified. Here, we estimate decadal frontal ablation rates on these glaciers using measurements of ice discharge and terminus change from 2020-2022. ... "

We have incorporated this suggestion into the abstract. We had to reword it somewhat to stay below the 150 abstract word limit.

Line 42: quantify the "remainder" too

Quantifying this part of sea level rise is tricky and not very useful because part of the terminus volume loss occurs underwater and thus should not be included in sea level rise calculations. Thus, only grounded termini that retreat will contribute to sea level rise and only the portion that is more than ~10% out of the water. We prefer to exclude this number here because it is so highly speculative. Similarly, glaciers that advance will contribute to sea level rise also as they

displace ocean water. For these reasons the volume of sea level rise due to terminus retreat and advance is likely negligible and highly uncertain.

Line 43: It would be helpful in the abstract if the ~3.6 mm of SLR was put into context of total global or barostatic or ice-driven SLR.

Please see first response above regarding comparing frontal ablation to other mass loss terms.

Line 81: “are remaining” -> ”remain”
done

Line 155: It may be personal ignorance, but I haven't seen “kinematics” used often in the literature. It's implicit here that this relates to terminus retreat, but perhaps this can be made more obvious for folks like myself by rewording, such as “Frontal ablation is the combination of two processes: a dynamic component related to variations in ice discharge, and a kinematic component related to variations in the terminus position. Quantifying the partition between these mass loss mechanisms is key to understanding underlying drivers of frontal ablation and highlighting where future efforts should be focused to reduce observational uncertainties.”

Agreed that what we want to show here is that terminus retreat and discharge are two very different processes, and you rightly point out that our original language was unclear. To better clarify this, we have renamed this section from ‘The kinematics and dynamics of frontal ablation’ to ‘Role of ice discharge vs terminus retreat’, and have removed references to kinematics and dynamics.

Line 169: Why is the ice cap not expected to surge again in the next few decades? Is there a citation for this?

We have rephrased this sentence to reference two sources that speak on this. Dowdeswell and Williams (1997) describe no surge-type glaciers in Severnaya Zemlya, while Willis et al. (2018) describe just this one event without evidence for past surging. We have thus updated the sentence as follows (line 201-202): “Vavilov Ice Cap is unlikely to surge again in the coming decades as no previous surge events have been identified since at least the 1980s, indicating a very long quiescent phase^{25,26}.”

Line 277: Please describe the division by 2 being necessary to account for average area more explicitly.

This is necessary because at the start of the decade no area was lost, while at the end of the decade 100% of the area was lost. Thus, on average, half the area was there for the decade, not the full area. We changed the language in the parenthesis here as follows (line 327): “division by 2 is necessary to account for the average area over the decade as no area was lost at the start of decade”

Line 280-297: How do you avoid sampling biases in the velocity measurements? E.g. some sensors provide observations in summer or winter only. Are seasonal variations in velocity enough to significantly change the results for a particular glacier?

We do not account for any potential seasonal bias in the velocity measurements, both because we do not know what this might be on a hemispheric basis, and because precise acquisition dates aren't provided for many of the velocity measurements. In particular, we rely on ITS_LIVE

derived velocities for many locations, but these are annual averages derived from a differing number of scenes with different acquisition dates for every glacier, with little of this information provided to the end user. We average velocities over a decade to reduce this potential effect, but unfortunately have no way to quantify it. Future work therefore needs to focus on better resolving seasonal changes in ice velocity and thus frontal ablation. When these observations are improved, we intend to ingest those observations and update our frontal ablation dataset. This is further elaborated upon in work by one co-author currently in review for the Eastern Arctic (Strozzi et al, in review), where they specifically addressed the representativeness of winter data with respect to mean annual value. They found that for non-surging glaciers short-term seasonal fluctuations in winter are relatively small and winter ice surface velocities are thus a good representative of mean annual velocities with an underestimation of less than 10%. Summer velocities, on the other hand, can be significantly larger than the annual mean with strong short-term seasonal fluctuations.

To directly address this in the text, we have added the following sentence (line 349):

“We do not account for seasonal variability in ice motion due to a lack of data to quantify this variability.”

Line 317: Where does the 10 come from?

Here, 10 is the minimum ice thickness. For more details, we cited Van Wychen et al. (2014) which better describes the equation. To avoid confusion, we also added the following sentence: “We use 10 m as a minimum ice thickness on the glacier margins.”

Line 364: It would be helpful to define climatic mass balance again here (the only other time its defined is in line 185). It would also be helpful to provide a few more sentences about how the model calculated this mass balance. The surface components are easier to understand with the provided information about melt, accumulation, and refreeze, but what is the model doing to calculate this “inside” or under the glacier? If nothing, then this term should to be referred to as SMB rather than climatic mass balance. If something, then info on the partitioning between these processes would shed some light on how the model is performing.

We added the definition of surface mass balance again here as follow:

“ \dot{B}_{clim} is defined as the mass changes due to snow accumulation, and melt at the surface and refreezing of melt and rain water below the surface¹. PyGEM assumes the basal mass balance is negligible.”

The term for additional processes happening inside and/or under the glacier would be described in basal-climatic mass balance (Cogley et al., 2011).

For more details on PYGEM, we cite Rounce et al. (2020) and direct the readers there.

Line 376: How does B_{clim} from the model compare to B_{clim} from the observations on the well-investigated glaciers? What does this tell us about the uncertainty of the B_{clim} estimates from the model?

Observations of the climatic mass balance of tidewater glaciers are sparse with measurements for only 5 tidewater glaciers (1 in Alaska, 1 in Arctic Canada South, and 3 in Svalbard). These observations are therefore far too limited to generate meaningful error metrics that could be used to quantify uncertainty of the climatic mass balance estimates. Instead, we have taken a rigorous approach to quantifying uncertainty associated with the climatic mass balance by running Monte Carlo simulations. For the reviewer’s knowledge, there are significantly more measurements of

the climatic mass balance on land-terminating glaciers that can be compared. For these glaciers, the mean absolute error associated with the annual glacier-wide mass balance is 0.54 m w.e. yr⁻¹. This is roughly comparable to the uncertainties associated with the estimate climatic mass balance, which have a mean±std of 0.36 ±0.31 m w.e. yr⁻¹. Given the limited observations of tidewater glaciers, we're confident that our approach reasonably reflects the uncertainty associated with the climatic mass balance estimates.

Reviewer #2 (Remarks to the Author)

Line 37: The fact that this study only assess Northern Hemisphere glaciers should be specified in the title.

We have added Northern Hemisphere to the title of the manuscript.

Line 40: It's worth noting that the increase in ablation between the 10-year periods (~ 4 Gt/yr is well within the error of 10 Gt/yr), despite the large glacier retreats & thinning.

Because we can only have 150 words in the abstract, there is no space to add a sentence here. We have, however, added this sentence on line 119: "While this suggests a slight increase in frontal ablation between the two decades, the estimates are within the uncertainties of each other."

Line 213-215: The basis for the conclusion "our first Northern Hemisphere wide frontal ablation estimate indicates that they play a critical role in global glacier mass balance through mass losses by frontal ablation" needs to be more clearly demonstrated. How does the ~50 Gt/yr of ice loss (which is within the error of the Huss and Hock (2015) estimate) compare to surface melt? (i.e. what's the % of accumulation lost to ablation)? In terms of sea level rise, 0.15 mm/yr is within the uncertainty of current total estimates.

Please see first response to reviewer 1 above regarding comparing frontal ablation to other mass loss terms.

line 153+: "Kinematics and Dynamics" It's not immediately clear what is specifically kinematics and dynamics refer to. My understanding is that kinematics and dynamics are two different ways to estimate a flux, with the former using continuity (e.g. mass balance) and the later using a description of the physical process (e.g. a flow model). Thus, either could describe advance/retreat or discharge (as far as I can tell, only kinematics are used in this paper). Please be more clear in your terminology here.

As described in the response to reviewer 1 on page 3, we have now removed the terms kinematics and dynamics to avoid any confusion about their meaning. Instead we now specifically describe short term and long term drivers of retreat, or the differences in terminus mass loss vs ice discharge in this section.

References cited

Ciraci, E., Velicogna, I. & Swenson, S. Continuity of the mass loss of the world's glaciers and ice caps from the GRACE and GRACE Follow-On missions. *Geophys. Res. Lett.* **47**, 9 (2020).

Cogley, J. G. et al. Glossary of Glacier Mass Balance and Related Terms. *IHP-VII Tech. Doc. Hydrol. No 86*, (2011).

- Hugonnet, R. et al. Accelerated global glacier mass loss in the early twenty-first century. *Nature* **592**, (2021).
- Noël, B. et al. Low elevation of Svalbard glaciers drives high mass loss variability. *Nat. Commun.* **11**, 1–8 (2020).
- Rounce, D. R., Hock, R. & Shean, D. E. Glacier Mass Change in High Mountain Asia Through 2100 Using the Open-Source Python Glacier Evolution Model (PyGEM). *Front. Earth Sci.* **7**, 1–20 (2020).
- Schuler, T. V et al. Reconciling Svalbard glacier mass balance. *Front. Earth Sci.* **8**, (2020).
- Strozzi, T., Wiesmann, A., Käab, A., Schellenberger, T. and Paul, F., 2022. Ice Surface Velocity in the Eastern Arctic from Historical Satellite SAR Data. *Earth System Science Data Discussions*, pp.1-42.
- Van Wychen, W. et al. Glacier velocities and dynamic ice discharge from the Queen Elizabeth Islands, Nunavut, Canada. *Geophys. Res. Lett.* **41**, 484–490 (2014).

Reviewers' Comments:

Reviewer #1:

Remarks to the Author:

Dear Nature Editors and Manuscript Authors,

In this revision by Kochtitzky et al, the authors have addressed many of the suggestions put forward by me and another anonymous reviewer which included some clarifications and a few grammatical changes for clarity.

My two main comments in the first draft pertained to 1) the assessment of these results in the context of global sea level rise, and 2) the utility of figure 3 versus other information regarding "climatic-basal" balance.

For 1), the authors have provided a detailed explanation pointing out some nuances (e.g. whether mass from below sea level is counted in the mass balance) that make it difficult to compare with some other studies. This explanation was informative, and some of these details would have been use to have in the manuscript. However, I understand that they may not fit within the typical condensed format of Nature articles, and perhaps these details are left to the reader.

For 2), the authors have pointed to their preference in keeping Figure 3, and provided some reasons why it is a useful figure in the main manuscript. Given the focus of the paper, I think this is a valid justification.

In response to this comment, however, I was a little confused that the authors indicated that climate-basal balance estimates for glaciers don't exist, making a comparison with frontal ablation impossible. I was initially under the impression that this component was actually calculated in this study – for example, line 87 indicates that a climatic mass balance model is used as component to estimate mean frontal ablation. In returning to the manuscript, I see that the climatic basal balance estimates were regional (Lines 218-242), rather than on a glacier-by-glacier basis. My comment here is somewhat related to the details provided to my point 1) above – a note about the different ways to partition mass balance would be useful in the manuscript. Also, an explicit note that the climate mass balance is not done for every glacier specifically would be useful for clarity.

The point about the climatic balance issue aside, the manuscript is comprehensive and clear, and I would recommend it for publication for its comprehensive assessment of frontal ablation on NH glaciers. As I indicated in my previous review, I think this paper will get cited often, both in the literature and in IPCC assessment reports.

Sincerely,

Michael Wood

We appreciate the reply to our comment from Dr. Wood. We have responded to each of the comment below in red and incorporated his suggesting into the text. The line numbers referenced below refer to the submitted document with track changes.

For 1), the authors have provided a detailed explanation pointing out some nuances (e.g. whether mass from below sea level is counted in the mass balance) that make it difficult to compare with some other studies. This explanation was informative, and some of these details would have been use to have in the manuscript. However, I understand that they may not fit within the typical condensed format of Nature articles, and perhaps these details are left to the reader.

To further clarify the relationship between frontal ablation and sea level rise, we have added the following to the end of the “sea-level rise calculations” subsection in the methods section (lines 417-419): “Estimates of mass loss below sea level are typically excluded from current global mass balance estimates⁹. Our frontal ablation estimates alone are not able to provide a total sea level rise estimate because frontal ablation is only one component of mass balance.”

For 2), the authors have pointed to their preference in keeping Figure 3, and provided some reasons why it is a useful figure in the main manuscript. Given the focus of the paper, I think this is a valid justification.

In response to this comment, however, I was a little confused that the authors indicated that climate-basal balance estimates for glaciers don't exist, making a comparison with frontal ablation impossible. I was initially under the impression that this component was actually calculated in this study ??? for example, line 87 indicates that a climatic mass balance model is used as component to estimate mean frontal ablation. In returning to the manuscript, I see that the climatic basal balance estimates were regional (Lines 218-242), rather than on a glacier-by-glacier basis. My comment here is somewhat related to the details provided to my point 1) above ??? a note about the different ways to partition mass balance would be useful in the manuscript. Also, an explicit note that the climate mass balance is not done for every glacier specifically would be useful for clarity.

We provide the definition of mass balance and the climatic-basal balance and frontal ablation components in the first sentence of the “Comparisons with regional mass balance” section. We refer the readers and reviewer to Cogley et al., 2011 for further reading on mass balance terms. We want to point out that the climatic balance and climatic-basal balance are two different terms. As mentioned in lines 67-82, and further explained in the subsection “Climatic mass balance below the flux gate” of the Methods Section, we indeed use a climatic balance model to account for the surface mass loss below the flux gate of the individual glaciers. On the other hand, in section “Comparisons with regional mass balance” we use our frontal ablation estimate, together with a regional geodetic mass balance (total balance) estimate, to derive the (seldom calculated) climatic-basal mass balance, which is a novel approach.

To further clarify that our climatic-basal balance estimates are done for each region, we have clarified the language on line 192 as follows: “to estimate the regional climatic-basal balance”.